# Can the Presence of Ovarian Corpus Luteum Modify the Hormonal Composition of Follicular Fluid in Mares?

**DOI:** 10.3390/ani10040646

**Published:** 2020-04-09

**Authors:** Katiuska Satué, Esterina Fazio, Pietro Medica

**Affiliations:** 1Department of Animal Medicine and Surgery, Faculty of Veterinary Medicine, CEU-Cardenal Herrera University, 46115 Valencia, Spain; 2Department of Veterinary Sciences, Veterinary Physiology Unit, Polo Universitario Annunziata, Messina University, 98168 Messina, Italy; fazio@unime.it (E.F.); pmedica@unime.it (P.M.)

**Keywords:** corpus luteum, cycling mares, follicular fluid, steroid hormones

## Abstract

**Simple Summary:**

The presence of corpus luteum (CL) in the ovary could exert a local differential effect on the hormonal composition of follicular fluid (FF), and it could also indirectly influence follicular development and oocyte quality. Using FF samples of follicles of different sizes in mares, we evaluated the differences between the composition of steroids (oestrogen, androgen and progesterone) in relation to the presence or absence of CL in the ovary. We suggest that FF concentrations of steroid hormones are not only related to follicular size but also to the presence or absence of CL in the ovary.

**Abstract:**

The hypothesis of this study was to investigate if the presence of corpus luteum (CL) in one ovary could modify the hormonal content of follicular fluid (FF) in the follicles. Sixty ovaries were taken after the slaughter of 30 clinically healthy mares. In relation to the sizes, the follicles were classified into three different categories, as small (20–30 mm), medium (31–40 mm) and large (≥41 mm). Blood samples were collected from the jugular vein of mares before their slaughter, and then the FF samplings were extracted from each single follicle. The ovaries that were collected were classified into two groups, according to the presence (CL-bearing) or absence (non-CL-bearing) of CL. The serum and FF samples were analysed for progesterone (P_4_), oestradiol-17β (E_2_), testosterone (T), androstenedione (A_4_) and dehydroepiandrosterone (DHEA). Intrafollicular P_4_ concentrations in large follicles of CL-bearing groups were lower than for non-CL-bearing ones. Intrafollicular E_2_ concentrations increased with the increase of the follicle diameter in both groups, CL-bearing and non-CL-bearing. However, in the FF with a large and medium follicle size, E_2_ concentrations were significantly higher in non-CL-bearing groups than in CL-bearing groups. T and A_4_ significantly increased in the large and medium follicle sizes when compared to the small follicle sizes in both groups, but higher concentrations in the non-CL-bearing group were obtained. Intrafollicular DHEA significantly decreased with the increase of the follicular diameter in both groups. Steroid hormones in FF dynamically changed, according to the presence or not of CL in the ovary. This study brings new knowledge on the role of the CL in the follicular hormonal composition in mares.

## 1. Introduction

During folliculogenesis, granulosa cells (GCs) are in constant communication with the oocyte, and the two cell types undergo a bidirectional nutrient transfer and paracrine signalling [1,2]. The intercellular communication is essential for follicular development, oocyte maturation and acquisition of follicular competence [3], related to the healthy intrafollicular environment [4]. Follicular fluid (FF) is produced during the last period of the secondary growing follicle and it remains closely connected to the cumulus-oocyte complexes (COC). FF is composed of both plasmatic exudates and GCs’ secretory pattern, also including the metabolic activity of thecal cells [5]. The FF contains metabolites [6,7], polysaccharides, proteins’ growth factors and hormones, which are locally synthetised and easily pass through the basal lamina to enter the antrum or escape toward circulating blood [8]. 

Numerous studies in different species clearly demonstrated that steroid hormones are essential for oocyte maturation and fertilisation, GC proliferation and differentiation, and eventual ovulation and luteinisation [2,3,4]. In mares, intrafollicular hormones, such as oestradiol-17β (E_2_), progesterone (P_4_) [6,9,10,11,12,13,14,15], androstenedione (A_4_) [12], testosterone (T) [13,14,15,16] and prostaglandin (PGF_2α_) [9,12], were documented. These hormones are related to the metabolic activity of ovarian cells, reflecting the physiological status of the follicle [8]. 

The estrous cycle in the mare integrates the follicular and luteal phases. The follicular phase is characterised by the presence of follicles at different stages of development and the simultaneous increase in the secretion of E_2_. Its duration is from 5 to 7 days, with variations from 3 to 9 days depending on the mare and the season; ovulation occurs approximately 24–48 hours before the end of the estrous cycle. The luteal phase is characterised by the presence of the CL, a fundamental structure in the regulation of the estrous cycle, supporting uterine development, embryonic implantation and the maintenance of pregnancy. The functional maturation of CL is characterised by a progressive increase in circulating P_4_ concentrations, structurally related to an increase in the CL diameter. P_4_ concentrations during the mild luteal phase (5 to 7 days) can reach between 4 and 10 ng/mL, and are correlated with a larger luteal diameter that remains elevated for a period of 6 to 10 days, and subsequently their concentrations decline. In the late right-handed (>13–16 days), PGF_2α_ is synthesised in the endometrium, reaches the ovary systemically and generates CL regression, characterised by a progressive decrease in P_4_ concentrations and luteal diameter [17].

The determination of the morphological characteristics of the ovary and CL in mares was used in recent years for clinical diagnostic practices and the application of assisted reproduction techniques [18]. Although the information on steroids in mares is extensive, it is unknown if the presence of CL in the ovary could modify the composition of steroid hormones in the FF. In cows [19], ewes [20] and camels [21], the composition of FF is modified according to the absence or presence of CL in the ovary. These differences could indicate the existence of a possible local effect of CL on the dynamic development of follicles. 

The aim of this study was to investigate if the existence of CL in one ovary could induce variations of intrafollicular P_4_, E_2_, T, A_4_ and DHEA concentrations in follicles of different sizes in mares. The results could bring new knowledge on the mechanisms involved in follicular development. Since these hormones are commonly used in culture medium as supplements for oocyte maturation procedures, they could be employed in assisted reproductive technology.

## 2. Materials and Methods 

### 2.1. Animals

All methods and procedures used in this study were in compliance with the guidelines of the Spanish law (RD 37/2014) that regulates the protection of animals at the time of slaughter and the EU directive (2010/63/EU) on the protection of animals used for scientific purposes. The Animal Ethics Committee for the Care and Use of Animals of the CEU-Cardenal Herrera University (Spain) concluded that the proposed study did not need ethical approval, as it did not qualify as an animal experiment under Spanish law.

To ensure the cyclic activity of the ovaries, the study was performed in the months of April and May 2018 of the breeding season in the northern hemisphere. The environmental temperature ranged between 27 °C and 31 °C, with 40%–60% relative humidity. A total of 30 clinically healthy mares, aged 6.6 ± 1.3 years, with a body weight paired to 533 ± 7.3 kg, were evaluated. In the livestock of origin, all animals were submitted to the same management conditions and feeding regime, including additional orchard grass-alfalfa mixed hay and free access to mineral salt and fresh water. Moreover, the official veterinarians for each livestock and slaughterhouse accepted the responsible participation in the present study, and only mares with a reproductive history of normality in the estrous cycles were studied. The veterinary examination of the animals before their slaughter consisted in verifying the official documentation, including livestock of origin, sanitary registration number, suitable health status, deworming and vaccination plan, and clinical and animal reproductive history.

### 2.2. Collection of Blood and Ovaries 

Before the animals’ slaughter, blood samples (20 mL) were collected from the jugular vein into heparinised tubes. Samplings were centrifuged at 1200 g for 10 minutes, and plasma aliquots were collected and stored at 4 °C using a portable cooler for the successive transportation to the laboratory for analysis. 

Post-mortem, the ovaries of all mares were collected; the time between the slaughter and the collection of the ovaries was under 2 hours, as reported by Hinrichs [22]. All ovaries were placed in containers added with 0.9% physiologic saline plus, penicillin (100 IU/mL) and streptomycin (50 mg/mL), and they were transported to the laboratory in individually labelled plastic bags in thermal containers (at 25 °C) [23].

### 2.3. Collection of Follicular Fluid 

The ovaries were washed three times with sterile saline; after this, all follicles were measured using the conventional callipers and were subsequently classified according to diameter: small (20–30 mm), medium (31–40 mm) and large (≥41 mm). On the basis of the presence or not of CL in the ovary, it was possible to distinguish two groups of ovaries: N.15 CL-bearing ovaries and N.15 non-CL-bearing ovaries. In each group, 10 small, 10 medium and 10 large follicles were analysed. 

In order to minimise the variations between the hormonal composition of FF, in CL-bearing ovaries the FF samplings were taken in the late-luteal phase because this is a period in which the functionality of the CL decreases. In this group, the FF of those follicles extracted from the ovaries whose CL sizes ranged from 30 to 40 mm were considered. The FF was aspirated using different sterile syringes and needles of 22G for each follicle. Following the collection, the FF samplings were centrifuged for 10 min at 1200 g to eliminate the cumulus oocyte complexes. Only the supernatant, represented by pure FF, was collected and stored in aliquots of 0.5 ml at −20 °C until analysis.

### 2.4. Determination of Progesterone (p4), Oestradiol-17 Beta (e2), Testosterone (t), Androstenedione (a4) and Dehydroepiandrosterone (DHEA) 

The intrafollicular and plasma concentrations of P_4_ (ng/mL) were determined using a solid-phase I-125 radioimmunoassay (RIA) (Coat-a-Count Progesterone, Diagnostic Products Co., Los Angeles, LA, USA). The minimal assay sensitivity of P_4_ was 0.1 ng/mL. The CV inter- and intra-assays for P_4_ were 16.1% and 4.3% at 3.5 ng/mL; 7.3% and 8.5% at 22.5 ng/mL; and 23.3% and 6.4% at 54.8 nmol/L. The concentrations of E_2_ (ng/mL) in FF and plasma were determined by a competitive enzyme-linked immunosorbent assay (E_2_ Sensitive, Demeditec ELISA DE4399) validated specifically for both types of fluids in the equine species, as previously reported [6]. The limit of detection of E_2_ was 1.4 ng/mL. The percentages of recovery in plasma and FF were 98.72% and 99.5%, respectively. The CVs’ intra- and inter-analyses at low and high concentrations were 7.87% and 5.52%, and 8.78% and 6.78%, respectively.

The concentrations of T (ng/mL), A_4_ (ng/mL) and DHEA (ng/mL) in plasma and FF were determined by enzyme immunoassay (EIA) techniques, using polyclonal antibodies (anti-T: R156, anti-A_4_: C9111 and anti-DHEA: C1011, Clinical Endocrinology Laboratory, UC Davis, California, USA), and hormone-peroxidase conjugates (conjugate of T: T3CMO-HRP, conjugate of A_4_: A3CMO-HRP and conjugate of DHEA: DHEA17CMO-HRP, Clinical Endocrinology Laboratory, UC Davis, California, USA), validated specifically in the equine species based on the technique proposed by Munro and Lasley [24] and previously used by these same researchers [25]. The detection limits for T, A_4_ and DHEA concentrations were 30 ng/mL, 25 ng/mL and 40 ng/mL, respectively. The CVs’ intra- and inter-assay for T, A_4_ and DHEA at high and low concentrations ranged between 5.9%–6.5% and 4.5%–6.9%, and 7.5%–9.9% and 7.5%–8.9%, respectively.

### 2.5. Statistical Analyses

Statistica 8.0 for Windows (Statsoft Inc., Bloomberg, NY, USA) was applied for the statistical analyses. The normality was calculated using the Kolmogorov Smirnov test to verify all data of different groups. A t-test was used to compare systemic and intrafollicular P_4_, E_2_, T, A_4_ and DHEA concentrations. The concentrations of each steroid hormone in the same follicle size categories were compared between CL-bearing and non-CL-bearing ovaries by ANOVA (One-way analysis of variance). A post-hoc comparison was carried out using Tukey’s test. A linear regression analysis was applied to evaluate the relationships among the hormonal concentrations; the correlations were expressed by Pearson’s correlation coefficient. Differences were considered statistically significant when *p* was paired to <0.05.

## 3. Results

The hormonal concentrations of systemic and FF hormones from the same follicle size categories, with CL-bearing and non-CL-bearing ovaries, are presented in Table 1. Systemic P_4_, E_2_, T, A_4_ and DHEA concentrations were lower than intrafollicular fluid (*p* < 0.05), except for FF values with a small size for non-CL-bearing ovaries. Systemic P_4_ concentrations for CL-bearing ovaries were higher than for non-CL-bearing ones (*p* < 0.05). Intrafollicular P_4_ concentrations in the large follicles of CL-bearing ovaries were higher than for non-CL-bearing ones (*p* < 0.05). Intrafollicular E_2_ concentrations in medium and large follicles of non-CL-bearing ovaries were higher than for CL-bearing ones (*p* < 0.05), and their concentrations increased with the increase of the follicle diameter in both groups (*p* < 0.05). T and A_4_ significantly increased for large and medium follicle sizes when compared to small follicle sizes in both the CL-bearing and non-CL-bearing groups (*p* < 0.05), with the highest concentrations in the non-CL-bearing group (*p* < 0.05). Intrafollicular DHEA concentrations decreased with an increase in the follicular diameter, with higher concentrations in large and medium follicle sizes than in small follicle sizes for both CL-bearing and non-CL-bearing groups (*p* < 0.05). The correlations of systemic and intrafollicular steroid hormones in both CL-bearing and non-CL-bearing ovaries are presented in Table 2. Positive and significative correlations were observed among E_2_ with T, A_4_ and DHEA (*p* < 0.01); A_4_ and T (*p* < 0.01); and DHEA with T and A_4_ (*p* < 0.01) concentrations. 

The follicle diameter was significantly and positively correlated with FF E_2_ (r = 0.84; *p* < 0.05), T (r = 0.72; *p* < 0.05) and A_4_ (r = 0.88; *p* < 0.05), respectively, and negatively correlated with DHEA (r = −0.80; *p* < 0.05).

## 4. Discussion

In the present study, intrafollicular and systemic P_4_, E_2_, T, A_4_ and DHEA concentrations obtained from the three categories of follicular sizes, in both CL-bearing and non-CL-bearing ovaries, were compared. The intrafollicular concentrations of these steroids were significantly higher than systemic ones. These differences remained for all follicle sizes and in both CL-bearing and non-CL-bearing ovaries. Significant positive correlations between systemic and intrafollicular concentrations of these steroids were found, as previously reported in cows for P_4_, E_2_ and T [18]. Moreover, FF provides a complex and peculiar environment for follicular growth, representing the metabolic and secretory activity of granulosa and thecal cells. Circumstantial evidence suggests that the evaluation of FF components, like steroid hormones, may provide additional information on their metabolic and dynamic changes, including into systemic concentrations [26,27].

Although these results could suggest that steroid hormones are synthesised in the follicular cells and are subsequently released into the systemic circulation, this evidence is subject to controversy. Indeed, although no correlations between the serum and FF of P_4_ and T in the preovulatory period were observed, FF P_4_ was positively correlated with serum in the postovulatory period [13]. On the contrary, Bøgh et al. [28] reported no relationship between systemic and FF P_4_ and E_2_ in transitional mares. However, these steroids are also produced by the adrenal cortex, and thus their presence in the blood does not directly reflect the gonadal steroidogenic capacity [29]. 

In the present study, no differences between the follicular diameters of CL-bearing and non-CL-bearing ovaries were obtained. During diestrus, in the presence of CL, multiple structures, such as small, medium antral and preovulatory follicles, can be present in the ovary of the mare [30]. Due to the great heterogenicity of follicle sizes that can occur even in mares for which the phase of the estrous cycle is unknown [31], one can presume not to expect significant differences between follicle sizes in CL-bearing and non-CL-bearing groups. However, positive correlations between the preovulatory follicles and CL diameters were documented [32]. A potential physiological mechanism that could explain this fact could be represented by the presence of an active and highly vascularised CL that may contribute to the better diffusion of growth factors and hormones throughout the ovarian stroma, favouring the quality of preantral follicles [31]. 

What is more, Ishak et al. [32] showed that the maximum preovulatory follicle diameter/maximum P_4_ concentration ratio was strongly correlated with the maximum CL diameter/maximum P_4_ concentration ratio. Moreover, positive correlations between the preovulatory follicles and CL diameters, and the capacity of the CL to produce P_4_, were found. In fact, the large follicles can continue to biosynthesise these steroids to the preovulatory state, as the results of this study suggest. 

As previously documented in mares [14,33,34,35], significant positive correlations between the diameter of follicles and E_2_ were shown in this study. Furthermore, E_2_ concentrations were significantly higher in large and medium ones than in small ones for both CL-bearing and non-CL-bearing groups. These results are in agreement with those reported in dromedary camels [21]. Moreover, a previous study revealed that cholesterol concentrations significantly increase in the follicles of greater vs. medium and smaller sizes, reflecting how the highest demand for this analyte is to increase the synthesis of steroids in this category of follicles [6]. 

Intrafollicular P_4_ concentrations in large follicles decrease in non-CL-bearing groups when compared with those obtained from small follicle sizes. However, P_4_ concentrations in the FF obtained from large follicles significantly decreased in CL-bearing groups when compared to those obtained from small and medium follicles in dromedary camels [21]. In non-CL-bearing ovaries, the higher concentration of P_4_ in large follicles when compared to small follicles suggests the luteinisation of GCs in dromedary camels [36]. In mares, Ishak et al. [32] reported that the diameter of the follicle influences the size of the subsequent CL and the P_4_ production. The larger and well-vascularised preovulatory follicles produce larger CLs, with a greater blood flow and, subsequently, a greater systemic P_4_ concentration in mares. However, although the results of the present study are in agreement with the increase in E_2_ and P_4_ along the follicular growth reported by Gérard et al. [37], they did not take into account the presence or absence of CL. On the contrary, Tsukada et al. [12] observed no differences in P_4_ concentrations in FF among follicles of different sizes. 

In the present study, high concentrations of E_2_ and T in non-CL-bearing mares were found. Beck et al. [16] classified the follicles in oestrogenic or non-oestrogenic follicles, when the FF content of E_2_ was > 40 or < 40 ng/mL, respectively. Compared to the non-oestrogenic follicles, the follicle diameter and E_2_ to T ratios were significantly greater in the oestrogenic ones, but no differences in the E_2_ to P_4_ ratios between both types of follicles were reported. These results partially confirm those shown in this study, since in both groups high concentrations of E_2_ and T were obtained, although P_4_ was lower in the non-CL-bearing group than in the CL-bearing group.

The E2 concentrations and the ratios of E_2_/T and/or P_4_ were significantly higher in large follicle sizes than in medium and small follicle sizes for the non-CL-bearing group, as previously reported in cows [19]. Kor et al. [19] suggested the existence of a possible local effect of the CL on such steroids. Thus, the results of the present study could suggest that intrafollicular E_2_ concentrations and E_2_ to T ratios could be used to evaluate the health status of the follicles, as previously reported in mares [16]. 

In other species such as cows, elevated E_2_ and E_2_/P_4_ ratios in FF indicated a more advanced stage of oocyte maturation and were associated with a higher chance of achieving pregnancy [38]. 

DHEA concentrations decreased in medium and large follicles with respect to small sizes for both CL-bearing and non-CL-bearing ovaries. Since DHEA is converted to T in ovarian connective tissue (theca/stroma) and is subsequently processed by the GCs to synthesise E_2_, it assumes the prohormone status of a predominant endogenous precursor and a metabolic intermediate in ovarian follicular steroidogenesis [39]. Thus, DHEA could be considered a key molecule at the crossroads, maintaining a critical balance between the androgen and oestrogen production. On the contrary, lower concentrations of DHEA in FF were detected, suggesting that the increase in circulating values during oestrus was caused by an extrafollicular source(s) [40]. Since previously reported concentrations were below the limit of the analytical assay, non-detectable concentrations or baseline samples from this study were not unexpected.

The biochemical FF composition of large follicles represents a significant marker of the metabolic and secretory pattern of follicular cells, with an applicative guide in formulating cell culture topics [35]. Since the maturation of the oocytes in the mare involves specific mechanisms of the species, some of which may be related to the steroids contained in the equine FF, it is of crucial importance to oocyte preparation for fertilisation and development [26,41]. The development of in vitro culture conditions able to mimic the maturation of the oocyte may help to improve embryo production. In fact, the use of FF as a means of maturation of equine oocytes increases the rate of fertilisation and the cleavage rate [42]. The pre-incubation of equine oocytes with FF or oviductal cells increases fertilisation rate [27,43]. In addition, preovulatory FF in which the maturation naturally occurs may sustain a better cytoplasmic maturation and thus a better competence for in vitro fertilisation (IVF) and development [44]. In bovines, the developmental potential of oocytes originating from CL-bearing and non-CL-bearing ovaries has been evaluated [45]. The percentage of blastocyst formation of oocytes originating from the medium and large follicles of non-CL-bearing ovaries was greater than that of oocytes originating from the small and medium follicles of CL-bearing ovaries. This pattern was the same in blastocysts per oocytes and blastocysts per cleaved embryos. Since blastocyst maturation is the crucial step for in vitro embryo production, it would be interesting to compare the possible role of steroids of FF in CL-bearing and non-CL-bearing ovaries as a medium of support in IVF programs, in order to improve the quality of maturation and developmental ovocitary competence. 

Intrafollicular P_4_ concentrations increase in mares with CL-bearing ovaries, while E_2_, A_4_ and T increase in non-CL-bearing ovaries without any modifications in DHEA during the follicular development. These different responses suggest that the presence of CL modifies the intrafollicular concentrations of these steroids. This information is not only important for enhancing the understanding of the follicular dynamics in mares but can also be helpful in assessing the developmental competence of follicles and CL, providing basic information for future clinical applications.

## 5. Conclusions

In conclusion, intrafollicular P_4_ concentrations increase in mares with CL-bearing ovaries, while E_2_, A_4_ and T increase in non-CL-bearing ovaries without any modifications in DHEA during the follicular development. These different responses suggest that the presence of CL modifies the intrafollicular concentrations of these steroids. This information is not only important for enhancing the understanding of the follicular dynamics in mares but can also be helpful in assessing the developmental competence of follicles and CL, providing basic information for future clinical applications.

## Figures and Tables

**Table 1 animals-10-00646-t001:** Means ± SD of systemic and intrafollicular steroid hormones in CL-bearing and non-CL-bearing ovaries.

Parameters	Intrafollicular Fluid				Intrafollicular Fluid			
	CL-Bearing Ovaries				Non-CL-Bearing Ovaries			
	Systemic	Small	Medium	Large	Systemic	Small	Medium	Large
N	15				15			
Follicles’ number		10	10	10		10	10	10
Follicles’ diameter (mm)		26.2 ± 2.31	36.0 ± 2.29	49.1 ± 5.03		26.7 ± 2.56	36.0 ± 1.77	50.7 ± 5.76
P_4_ (ng/mL)	3.12 ± 0.21 ^AB^	10.45 ± 1.85	7.47 ± 0.75	18.8 ± 0.93 ^B^	2.65 ± 0.45	3.34 ± 1.90	6.80 ± 0.23 ^D^	9.10 ± 2.12 ^D^
E_2_ (pg/mL)	73.60 ±13.31 ^A^	652.9 ± 241.3	1484.6 ± 181.8 ^C^	1542.1 ± 113.6 ^C^	87.12 ± 19.73 ^A^	833.6 ± 231.3	1532.1 ± 270.6 ^B,C^	1773.8 ± 85.79 ^B,C^
T (ng/mL)	0.55 ± 0.19 ^A^	1.97 ± 0.17	3.63 ± 0.87 ^C^	5.76 ± 1.35^C^	0.39 ± 0.13 ^A^	1.14 ± 0.20	4.55 ± 0,75 ^B,C^	10.9 ± 2.56 ^B,C^
A_4_ (ng/mL)	1.50 ± 0.79 ^A^	60.1 ± 12.5	140.1 ± 24.2 ^C^	241.5 ± 28.5 ^C^	1.95 ± 0.96 ^A^	72.52 ± 13.18	201.4 ± 35.4 ^B,C^	274.3 ± 23.2 ^B,C^
DHEA (ng/mL)	4.06 ± 1.71 ^A^	14.5 ± 2.31 ^C^	7.18 ± 1.71 ^C^	4.36 ± 0.71 ^C^	4.38 ± 1.64 ^A^	14.4 ± 2.99	7.55 ± 1.72 ^C^	4.38 ± 1.01 ^C^

P_4_: progesterone; E_2_: oestradiol-17β; T: testosterone; A_4_: androstenedione; DHEA: dehydroepiandrosterone. Different superscripts indicate significant differences: A = *versus* intrafollicular fluid (*p* < 0.05); B = *versus* non-CL-bearing ovaries (*p* < 0.05); C= *versus* small follicles (*p* < 0.05); D = *versus* systemic (*p* < 0.05).

**Table 2 animals-10-00646-t002:** Correlations of systemic and intrafollicular steroid hormones in CL-bearing and non-CL-bearing ovaries.

Parameters	P_4_ (ng/mL)	E_2_ (ng/mL)	A_4_ (ng/mL)	DHEA (ng/mL)
T (ng/mL)	non-CL-bearing (r = 0.33) CL-bearing (r = 0.34)	non-CL-bearing (r = 0.70; *p* < 0.01) CL-bearing (r = 0.72; *p* < 0.01)	non-CL-bearing (r = 0.81 *p* < 0.01) CL-bearing (r = 0.79 *p* < 0.01)	non-CL-bearing (r = −0.79; *p* < 0.01) CL-bearing (r = −0.76; *p* < 0.01)
A_4_ (ng/mL)		CL-bearing (r = 0.80; *p* < 0.01)non-CL-bearing (r = 0.85; *p* < 0.01)		CL-bearing (r = −0.82; *p* < 0.01)non-CL-bearing (r = −0.89; *p* < 0.01)
DHEA (ng/mL)		CL-bearing (r = −0.83; *p* < 0.01)non-CL-bearing (r = −0.71; *p* < 0.01)

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
