# Peer review of "Can the Presence of Ovarian Corpus Luteum Modify the Hormonal Composition of Follicular Fluid in Mares?"

_animals, 2020, doi:10.3390/ani10040646_

Round 1

Reviewer 1 Report

This study was designed to analyze the effect of the corpus luteum presence on the hormonal content of the follicular fluid. The manuscript is well written, and authors have used the right experimental approach to answer all the questions in this study. Results support most of the findings described in the discussion section. However, there are few scientific and grammatical issues need to be addressed:

The goal of this study was to determine the effects of corpus luteum presence on hormonal content of the follicular fluid. However, several findings described in the discussion are focused on hormonal content in small vs large follicles which have been already reported in previous publications cited in this manuscript (for example: line 217-222). Authors need to focus more on the direct comparison of CL vs non-CL follicular fluid hormonal level which are the actual novel aspects of this study.

Line 200: “Intrafollicular P4 concentrations in large follicles decrease in non-CL-bearing compared with those obtained from small follicle sizes” seems contradictory to the reference cited by the authors. In the reference article cited by author P4 concentration is higher in large follicles compared to the small follicles. Please explain.

In the introduction, the authors described that findings from this study can help design the culture medium for oocyte maturation procedures. However, the authors has not provided any explanation in the discussion section about the strategies of using findings obtained in this study for oocyte maturation media. Please add a few lines about how authors proposed to use this information for oocyte maturation media optimization.

Line 121: Please expand the “EIA”. Any abbreviations used in this manuscript needs to be described while using the first time.

Line 207: Gerald et al (observed) no difference (please insert the word observed)

Line 225: “According to the last author” please be clear about what does “last authors” mean. Cite reference or

Author Response

Reviewer #1:

This study was designed to analyze the effect of the corpus luteum presence on the hormonal content of the follicular fluid. The manuscript is well written, and authors have used the right experimental approach to answer all the questions in this study. Results support most of the findings described in the discussion section. However, there are few scientific and grammatical issues need to be addressed:

The goal of this study was to determine the effects of corpus luteum presence on hormonal content of the follicular fluid. However, several findings described in the discussion are focused on hormonal content in small vs large follicles which have been already reported in previous publications cited in this manuscript (for example: line 217-222). Authors need to focus more on the direct comparison of CL vs non-CL follicular fluid hormonal level which are the actual novel aspects of this study.

Response = The authors have now compared directly the hormonal content of the FF of mares’ovaries carrying CL vs non CL bearing. Therefore, other comparisons regarding follicular size as small versus large follicles (Sirois et al., Watson and Sertich, and Watson et al.), without considering the presence or absence of CL have been eliminated as reviewer's request.

Line 200: “Intrafollicular P4 concentrations in large follicles decrease in non-CL-bearing compared with those obtained from small follicle sizes” seems contradictory to the reference cited by the authors. In the reference article cited by author P4 concentration is higher in large follicles compared to the small follicles. Please explain.

Response = The authors have verified the results of the El-Shahat et al. (2019) [33] study:

P4 concentrations were lower in large follicle in CL-bearing (21.62 ± 6.59) than non CL-bearing (31.51 ± 0.74).

Now, the authors have revised the following paragraph:

“Intrafollicular P4 concentrations in large follicles decrease in non CL-bearing compared with those obtained from small follicle sizes. However, P4 concentrations in the FF obtained from large follicles significantly decreased in CL-bearing as compared to those obtained from small and medium follicles in dromedary camels [20]. The higher concentrations of P4 in large compared to small follicles in non CL-bearing ovaries suggest the luteinization of GCs in dromedary camels (El-Shahat et al., 2019 [33]). In mares, Ishak et al. 2917 [29] reported that the diameter of the follicle influences the size of the subsequent CL and the P4 production. The larger and well-vascularized preovulatory follicles produce larger CLs, with greater blood flow and, subsequently, greater systemic P4 concentration in mares.

In the introduction, the authors described that findings from this study can help design the culture medium for oocyte maturation procedures. However, the authors has not provided any explanation in the discussion section about the strategies of using findings obtained in this study for oocyte maturation media. Please add a few lines about how authors proposed to use this information for oocyte maturation media optimization.

Response = The information related to oocyte maturation media optimizationis reflected in the following paragraph: “Developmental potential of oocytes originating from ovaries CL-bearing and non CL-bearing ovaries in bovine (Shabankareh et al. [45]) has been evaluated. The percentage of blastocyst formation of oocytes originating from medium and large follicle of ovaries non CL-bearing was greater than that of oocytes originating from small and medium follicle of ovaries CL-bearing. This pattern was the same in blastocyst per oocytes and blastocyst per cleaved embryos. Since that blastocyst maturation is the crucial step for in vitro embryo production, it would be interesting to compare the possible role of steroid del FF in CL-bearing and non CL-bearing as medium of support in IVF programs in order to improve the quality of maturation and developmental competence ovocitary”. This information has been now added at request of reviewer.

Line 121: Please expand the “EIA”. Any abbreviations used in this manuscript needs to be described while using the first time.

Response = Expansions of abbreviation “enzyme immunoassay (EIA) and in vitro fertilization (IFV)” have been described, as the reviewer suggests.

Line 207: Gerald et al (observed) no difference (please insert the word observed)

Response = The authors have inserted the word “observed” in Tsukada et al. (2008) [12] “observed” no differences in P4 concentrations in FF among follicles of different sizes.

Line 225: “According to the last author” please be clear about what does “last authors” mean. Cite reference or

Response = “According to the last author” the sentence was changed according to the reference “Kor et al. 2013 [18]” at request of review.

Reviewer 2 Report

The authors tried to find any correlation between CL or non-CL ovaries and several hormones related to estrus. I believe that the authors should elaoborate in the Introduction section the role of CL in mares. If so, they will find that the most of the correlations found and in this study exist. Secondly, the number of casees used is relative low for any conclusions. Third, why the authors used 3 groups of diameter of follicles and why 2 or 4 groups. In case they will use 2 groups the results may be different.

Author Response

The authors tried to find any correlation between CL or non-CL ovaries and several hormones related to estrus. I believe that the authors should elaborate in the introduction section the role of CL in mares. If so, they will find that the most of the correlations found and in this study exist. Secondly, the number of cases used is relative low for any conclusions. Third, why the authors used 3 groups of diameter of follicles and why 2 or 4 groups. In case they will use 2 groups the results may be different.

Response = The role of CL in mares have been introduced. Correlations have been verified. The authors used three groups in order to homogeneize the different sizes of follicles obtained. Furthemore, similar criteria for classifying in slaughterhouse ovaries were adopted by Tsukada et al. (2008) and by these same authors, Satué et al. (2019,a,b) in previous research on mares.

  1. Satué, K.; Fazio, E.; Ferlazzo, A.; Medica, P. Intrafollicular and systemic serotonin, oestradiol and progesterone concentrations in cycling mares. Reprod. Dom. Anim. 2019, 54, 1411–1418.
  2. Satué, K.; Fazio, E.; Ferlazzo, A.; Medica, P. Hematochemical patterns in follicular fluid and blood stream in cycling mares: a comparative note. J. Equine Vet. Sci. 2019, 80, 20–26.
  3. Tsukada, T.; Kojima, A.Y.; Sato, K.; Moriyoshi, M.; Koyago, M.; Sawamukai, Y. Intrafollicular concentrations of steroid hormones and PGF2α inrelation to follicular development in the mares during the breeding season. J. Equine Sci. 2008, 19, 31–34.

Reviewer 3 Report

The manuscript by Satué et al. (animals-754145) aimed to determine whether the presence of corpus luteum could modify the hormonal content of follicular fluid in mares. The subject of the manuscript is of interest and in need of research. The manuscript is relatively well written. However, the manuscript requires major revision before the publication in Animals. The list of specific comments that also should be addressed are defined as follows:

  1. Introduction: additional information about equine estrous cycle should be added. Since “the CL of the mare is functional for about fourteen to fifteen days of the cycle” (lines 57), if all classes of ovarian follicles are presented during this time?
  2. What was the day of estrous cycle (what stage of luteal phase) of animals in the examined groups? Was it the same for all animals within group? It is important since early-luteal, mid-luteal or late-luteal CLs may modify the hormonal composition of follicular fluid in different way.
  3. What was the rationale for DHEA measurement.
  4. If atretic follicles were observed? How they were excluded from the experiment?
  5. Line 127: should be “Statsoft” instead of “Statsowft”
  6. It is worth mentioning in the discussion that steroids are also produced by adrenal cortex, thus their presence in the blood does not directly reflect gonadal steroidogenic capacity.
  7. Sometimes the Discussion section is written in the style of a review paper, and it should be corrected. Each paragraph in this section should end with a concluding statement.

Author Response

The manuscript by Satué et al. (animals-754145) aimed to determine whether the presence of corpus luteum could modify the hormonal content of follicular fluid in mares. The subject of the manuscript is of interest and in need of research. The manuscript is relatively well written. However, the manuscript requires major revision before the publication in Animals. The list of specific comments that also should be addressed are defined as follows:

Introduction: additional information about equine estrous cycle should be added. Since “the CL of the mare is functional for about fourteen to fifteen days of the cycle” (lines 57), if all classes of ovarian follicles are presented during this time?.

What was the day of estrous cycle (what stage of luteal phase) of animals in the examined groups? Was it the same for all animals within group? It is important since early-luteal, mid-luteal or late-luteal CLs may modify the hormonal composition of follicular fluid in different way.

Response = In order to minimize variations between results, samples of FF were taken in late-luteal phase.

What was the rationale for DHEA measurement.

Response = Since DHEA is converted to T in ovarian connective tissue (theca/stroma) and is subsequently processed by the GCs to synthetize E2, it assumes the prohormone status of a predominant endogenous precursor and a metabolic intermediate in ovarian follicular steroidogenesis. So, DHEA could be considered a key molecule at the crossroads, maintaining a critical balance between androgen and oestrogen production. The authors send to reviewer the rationale for all androgens used in the study.

The concentrations of T (ng/ml), A4 (ng/ml) and DHEA (ng/ml) were determined by EIA techniques, using polyclonal antibodies (anti-T: R156, anti.A4: C9111 and anti-DHEA: C1011), and hormone-peroxidase conjugates (conjugate of T: T3CMO-HRP, conjugate of A4: A3CMO-HRP and conjugate of DHEA: DHEA17CMO-HRP) obtained and characterized in the Department of Animal Physiology of the Faculty of Veterinary of Complutense University of Madrid. The validation of the parameters of the EIA technique for T, A4 and DHEA in equine serum was based on the technique previously proposed by Munro and Lasley (1988), with the dilution of the antibody in carbonate / bicarbonate buffer until reaching the corresponding dilution, the wells were then covered with 100 μl except well A1 which was left blank.

After sealing and incubation of the plates at 4 °C for 16 hours, they were washed three times with wash solution (200 μl per well) to remove excess antibody that does not bind to the plate. The reaction occurred between the free hormone, either the sample or standard, and the hormone conjugated to the enzyme. The conjugate was diluted in EIA buffer solution. Then, the reaction proceed to dilute the samples in the conjugate solution (50 μl) in 250 μl of conjugate, homogenizing the sample carefully by means of an agitator (Reax 2,000, Heindolph). 60 μl of this solution were used together with 40 μl of EIA buffer to cover the wells of the polystyrene plate (Biogreiner). For the standard curve, the wells were covered with 50 μl of each of the standards together with 50 μl of EIA buffer. The standard and problem samples were determined in duplicate. To calculate the highest antibody binding wells, four wells of column A were upholstered with 50 μl of the conjugate solution, together with 50 μl of EIA buffer. The plates were sealed (ICN Biomedical Inc.), incubated for a period of 2 hours at room temperature. Next, the fractions of free and bound hormone bound to the adsorbed antibodies in the solid phase were separated by dumping the plates and subsequent washing with 200 μl of washing solution per well.

100 μl of tetramethylbenzidine (TMB, Neogen, USA) was added to all wells of the plate, incubated for 20 minutes at room temperatura for colour development by the chromogen and the addition of 100 μl of braking solution. Once the substrate reaction wass topped, the optical density of the color developed by means of anautomatic EIA reader (Bio-Tek Instruments) wasread, which, using 450 and 600 nm filters, performed a bichromatic reading, eliminating the colour produced by a possible non-specificbackgroundreaction. Hormonal concentrationswerecalculatedbyusing software developed for the competition EIA technique (ELISA-AID Eurogenetics, Belgium). The dose-response curve was plotted against the percentage of binding of the hormone without labeling the antibody with the different standard dilutions of T, A4 and DHEA.

The detection limits of the technique for T, A4 and DHEA concentrations were 30 pg / ml, 25 pg / ml and 40 pg / ml, respectively. The percentages of recovery were 95% and 98%, at high and low concentrations, respectively. The intra-analysis CVs for A4 and T ranged between 5.9-6.5% and 4.5-6.9% at high and low concentrations, respectively. Likewise, the inter-analysis CV ranged between 8.9-7.5% and 7.5-9.9% for A4 and T, respectively. Although the polyclonal anti-T R 156 antibody showed cross-reactions with 5-α-DHT (20%), 5-β-DHT (5.0%), A4 (11.5%), andostenediol (3.5%), androstenolone (3.21%), epitestosterone (0.10%), with estradiol, progesterone (P4) and cortisol, it was less than 1%. The anti-A4 polyclonal antibody showed cross-reactions with T (3.16%), oestrone sulphate (2%), estradiol (2.12%), although it was less than 1% with P4 and cortisol. Although the anti-DHEA C1011 polyclonal antibody showed cross-reactionswith DHEA-hemisuccinate (16.0%), DHEA-sulphate (12.6%) and A4 (6.0%), was less than 1% with T, estradiol, P4 and cortisol. The linearity of the technique was demonstrated by serial dilutions of a set of FF and serum samples in a range between 1: 1 and 1:32 (1:1, 1:2, 1:4, 1:8, 1:16 y 1:32). T and DHEA is linear up to 1:16 and A4 and A4 up to 1: 8.

If atretic follicles were observed? How they were excluded from the experiment?

Response = In our study, only follicles with a size equal to or greater than 20 mm non-atretic were used.

Line 127: should be “Statsoft” instead of “Statsowft”

Response = This mistake has been corrected.

It is worth mentioning in the discussion that steroids are also produced by adrenal cortex, thus their presence in the blood does not directly reflect gonadal steroidogenic capacity.

Sometimes the Discussion section is written in the style of a review paper, and it should be corrected. Each paragraph in this section should end with a concluding statement. 

Response = The sentence “Steroids are also produced by adrenal cortex, thus their presence in the blood does not directly reflect gonadal steroidogenic capacity (Watson and Hinrichs, 1989)” has been added in the discussion section. Concluding statement in each paragraph has been included.

Reviewer 4 Report

please see attached file.

Author Response

Reviewer #4:

1.-It is suggested to describe the CL size in the Materials and Methods.

Response = The CL size is now added in M & M section.

2.-The sensitivities and CVs of inter- and intra-assays of hormone assays are suggested to be listed in the Materials and Methods, although they were validated in previous study.

Response = The authors validated the techniques used in a previous investigation in this same species (Satué et al. [24])

  1. Satué, K.; Marcilla, M.; Medica, P.; Ferlazzo, A.; Fazio, E. Testosterone, androstenedione and dehydroepiandrosterone concentrations in pregnant Spanish Purebred mare. Theriogenology 2019, 123, 62-67.

3.-Table 1 should be checked again and improved. Is the systemic P4 level 18.8 ng/ml? If it is, really does no difference exist between CL bearing and non CL bearing groups? In the first column (stub), the first two cells please fill “number of follicle” and “follicle diameter”, respectively. The “a” and “B” should be changed to superscripts. The abbreviations in the stubs should be defined in the footnote. The significances of differences should be check again carefully, especially the E2 levels in medium and large follicles and A4 levels in large follicles between CL bearing and non CL.

Response = The systemic P4 level is paired to 18.8 ng/ml in non CL bearing group, and higher than systemic P4 levels of CL bearing group (paired to 3.12 ng/ml). The numbers N in each group and the measurements of the follicular diameters (mm) have been introduced in the first column. The abbreviations in the stubs should be defined in the footnote. Now, superscript have been explained. The authors have been verified the significances of E2 levels in medium and large follicles and A4 levels in large follicles between CL bearing and non CL.

4.-The significances of the correlations in Table 2 should be illustrated.

Response = The significances of the correlations in Table 2 have been illustrated.

5.-L146: “higher” change to “lower”? Please check again.

Response = “Higher” have been changed by “lower” at request of reviewer.

6.-Many errors exist in the manuscript, please check again or seek help for English editing. For examples:

(1) L24-25: “slaughter and then,” change to “slaughter, and then”

(2) L35: “change” change to “changed”

(3) “reproductive animal history” change to “animal reproductive history”

(4) L167: “granulose” change to “granulosa”

(5) L178: “diestrous” change to “diestrus”

(6) L207: “they” change to “who”

(7) L239: “Since that” change to “Since”

Response = All these changes have been realyzed, as the reviewer proposed.

Round 2

Reviewer 2 Report

The authors made a small improvement on their revision. They provided just the updated references regarding the role of hormones and CL. They did not elaborate more on this topic and they did not even justified the number of groups they used or even the number of cases used, as I expressed in the previous revision

Author Response

REVIEWER

The authors made a small improvement on their revision. They provided just the updated references regarding the role of hormones and CL. They did not elaborate more on this topic and they did not even justified the number of groups they used or even the number of cases used, as I expressed in the previous revision.

Authors = The revision of estrous cycle, including phases, present structures and hormonal dynamics, has been expressed in the introductory section, as the reviewer has suggested.

“The estrous cycle in the mare integrates the follicular and luteal phases. The follicular phase is characterized by the presence of follicles at different stages of development and the simultaneous increase in the secretion of E2. Its duration is from 5 to 7 days, with variations from 3 to 9 days depending on the mare and the season; ovulation occurs approximately 24-48 hours before ending of estrous. The luteal phase is characterized by the presence of the CL, a fundamental structure in the regulation of the estrous cycle, supporting uterine development, embryonic implantation and the maintenance of pregnancy. The functional maturation of CL is characterized by a progressive increase in circulating P4 concentrations, structurally related to an increase in CL diameter. P4 concentrations during the mild luteal phase (5 to 7 days) can reach between 4 and 10 ng / ml, and are correlated with a larger luteal diameter, remaining elevated for a period of 6 to 10 days, and subsequently their concentrations decline. In the late right-handed (> 13-16 days), PGF2α is synthesized in the endometrium, reaches the ovary systemically, and generates CL regression, characterized by a progressive decrease in P4 concentrations and luteal diameter [17]”.

We employ two groups, CL-bearing and non CL-bearing. Within each of these groups, we consider in turn three subtypes, depending on the follicular diameters: small, medium and large sizes. In any case, the authors tried to homogenize the number of follicles and samples of FF in each group in order to compare them correctly. Specifically, in the CL-bearing group, these authors consider the presence of CL, whose size varied in any case between 30 and 40 mm. After checking for the presence of CL, we classified with the follicles present.

Reviewer 3 Report

Revision improved this manuscript substantially. However, as I still have some comments, the manuscript requires revision before the publication in Animals. The list of specific comments that should be addressed are defined as follows:

  1. Introduction: additional information about equine estrous cycle should be added (length of estrous cycle, phases of estrous cycle etc.)
  2. Line 109 “…FF samples in CL-bearing ovaries were taken in late-luteal phase…”: Since CL begins to regress during the late luteal phase, the major concern is the rationale for choosing this stage of estrous cycle. Why not mid-luteal phase (with fully functional CL)? It should be explain in the aim of study or material and method section of the manuscript

Author Response

Revision improved this manuscript substantially. However, as I still have some comments, the manuscript requires revision before the publication in Animals. The list of specific comments that should be addressed are defined as follows:

Introduction: additional information about equine estrous cycle should be added (length of estrous cycle, phases of estrous cycle etc.).

Authors =Additional information about equine estrous cycle has been added at request of reviewer.

Line 109 “…FF samples in CL-bearing ovaries were taken in late-luteal phase…”: Since CL begins to regress during the late luteal phase, the major concern is the rationale for choosing this stage of estrous cycle. Why not mid-luteal phase (with fully functional CL)? It should be explain in the aim of study or material and method section of the manuscript.

Authors =These authors apologize to the reviewer for the mistake made in Table 1. The systemic P4 concentrations of non CL-bearing (the value of 18.8 was corrected with 2.65 ng/ml) and of small follicle size (the value of 10.77 was corrected with 3.34 ng/ml) were the lowest! The comparison of results obtained for P4 concentrations with published data reported for nonpregnant mares: <1 ng/mL during estrous,1.5–2.5 ng/mL at 24 hours postovulation, 2–5 ng/mL at 48 hours postovulation, and 8–20 ng/mL at 5–8 post-ovulation days (Panzani et al., 2017), could indicate mid-luteal phase. Nevertheless, from an interpretation of macroscopic aspect of CL, on the basis of its diameter and the presence of three different follicular sizes, according to the systemic P4 values, it is possible to presume that, exactly, the samples were taken at late way through the luteal phase, because it is a period in which the functionality of the CL is decreasing. Hence, the systemic E2 concentrations could also corroborate this hypothesis.

Panzani D, Di Vita M, Lainé AL, Guillaume D, Rota A, Tesi M, Vannozzi I, Camillo F, Corpus luteum vascularization and progesterone production in autumn and winter cycles of the mare: relationship between ultrasonographic characteristics of corpora lutea and plasma progesterone concentration in the last cycles before anestrus., Journal of Equine Veterinary Science (2017), 56, 35-39.

In the introductory section, the authors have clearly described the phases of the estrous cycle, the dominant structures in the ovary and the hormonal dynamics present in each one in the mare.

“The estrous cycle in the mare integrates the follicular and luteal phases. The follicular phase is characterized by the presence of follicles at different stages of development and the simultaneous increase in the secretion of E2. Its duration is from 5 to 7 days, with variations from 3 to 9 days depending on the mare and the season; ovulation occurs approximately 24-48 hours before ending of estrous. The luteal phase is characterized by the presence of the CL, a fundamental structure in the regulation of the estrous cycle, supporting uterine development, embryonic implantation and the maintenance of pregnancy. The functional maturation of CL is characterized by a progressive increase in circulating P4 concentrations, structurally related to an increase in CL diameter. P4 concentrations during the mild luteal phase (5 to 7 days) can reach between 4 and 10 ng / ml, and are correlated with a larger luteal diameter, remaining elevated for a period of 6 to 10 days, and subsequently their concentrations decline. In the late right-handed (> 13-16 days), PGF2α is synthesized in the endometrium, reaches the ovary systemically, and generates CL regression, characterized by a progressive decrease in P4 concentrations and luteal diameter [17]”.

In this study, the authors considered the FF of those follicles extracted from the ovaries whose CL sizes ranged from 30 to 40 mm. This information has been added in methods section.
